Building a synthesis of economic costs of biological invasions in New Zealand

Bodey Thomas W. 1 2
Carter Zachary T. zach.carter@auckland.ac.nz 1
Haubrock Phillip J. 3 4
Cuthbert Ross N. 5 6
Welsh Melissa J. 7
Diagne Christophe 8
Courchamp Franck 8
1 School of Biological Sciences, University of Auckland , Auckland , New Zealand
2 School of Biological Sciences, University of Aberdeen , Aberdeen , United Kingdom
3 Department of River Ecology and Conservation, Senckenberg Research Institute and Natural History Museum Frankfurt , Gelnhausen , Germany
4 Faculty of Fisheries and Protection of Waters, University of South Bohemia , České Budějovice , Czech Republic
5 GEOMAR Helmholtz-Zentrum für Ozeanforschung Kiel , Kiel , Germany
6 School of Biological Sciences, The Queen’s University Belfast , Belfast , United Kingdom
7 Scion , Christchurch , New Zealand
8 CNRS, AgroParisTech, Ecologie Systématique Evolution, Université Paris-Saclay , Orsay , France
Li Chenxi
Electronic publication date: 2022 Aug 15
Publication date: 2022
Volume: 10
Electronic Location ID: e13580
Received 2022 Jan 17; Accepted 2022 May 22
Copyright: ©2022 Bodey et al.
Copyright year: 2022
Copyright holder: Bodey et al.
License: This is an open access article distributed under the terms of the Creative Commons Attribution License, which permits unrestricted use, distribution, reproduction and adaptation in any medium and for any purpose provided that it is properly attributed. For attribution, the original author(s), title, publication source (PeerJ) and either DOI or URL of the article must be cited.
License URL: https://creativecommons.org/licenses/by/4.0/

Keywords: Biosecurity, Eradication, Invasive alien species, InvaCost, Island, Monetary impacts, Resource damages and losses, Socioeconomic sectors

Funding: French National Research Agency ANR-14-CE02-0021 BNP-Paribas Foundation Climate Initiative AXA Research Fund Chair of Invasion Biology BiodivERsA and Belmont-Forum European Union’s Horizon 2020 Marie Skłodowska-Curie Fellowship 747120 New Zealand International Doctoral Research scholarship Alexander von Humboldt Foundation BiodivERsA-Belmont Forum Project “Alien Scenarios” BMBF/PT DLR 01LC1807C This work was funded by the French National Research Agency (ANR-14-CE02-0021) and the BNP-Paribas Foundation Climate Initiative for funding the InvaCost project that allowed the construction of the InvaCost database. The present work was conducted following a workshop funded by the AXA Research Fund Chair of Invasion Biology and is part of the AlienScenario project funded by BiodivERsA and Belmont-Forum call 2018 on biodiversity scenarios. Thomas W. Bodey was funded by European Union’s Horizon 2020 research and innovation programme Marie Skłodowska-Curie fellowship (Grant No. 747120). Zachary T. Carter was funded by a New Zealand International Doctoral Research scholarship. Ross N. Cuthbert was funded by Humboldt Fellowship funding from the Alexander von Humboldt Foundation. Christophe Diagne was funded by the BiodivERsA-Belmont Forum Project “Alien Scenarios” (BMBF/PT DLR 01LC1807C). The funders had no role in study design, data collection and analysis, decision to publish, or preparation of the manuscript.

==============================
Biological invasions are a major component of anthropogenic environmental change, incurring substantial economic costs across all sectors of society and ecosystems. There have been recent syntheses of costs for a number of countries using the newly compiled InvaCost database, but New Zealand—a country renowned for its approach to invasive species management—has so far not been examined. Here we analyse reported economic damage and management costs incurred by biological invasions in New Zealand from 1968 to 2020. In total, US$69 billion (NZ$97 billion) is currently reported over this ∼50-year period, with approximately US$9 billion of this considered highly reliable, observed (c.f. projected) costs. Most (82%) of these observed economic costs are associated with damage, with comparatively little invested in management (18%). Reported costs are increasing over time, with damage averaging US$120 million per year and exceeding management expenditure in all decades. Where specified, most reported costs are from terrestrial plants and animals, with damages principally borne by primary industries such as agriculture and forestry. Management costs are more often associated with interventions by authorities and stakeholders. Relative to other countries present in the InvaCost database, New Zealand was found to spend considerably more than expected from its Gross Domestic Product on pre- and post-invasion management costs. However, some known ecologically (c.f. economically) impactful invasive species are notably absent from estimated damage costs, and management costs are not reported for a number of game animals and agricultural pathogens. Given these gaps for known and potentially damaging invaders, we urge improved cost reporting at the national scale, including improving public accessibility through increased access and digitisation of records, particularly in overlooked socioeconomic sectors and habitats. This also further highlights the importance of investment in management to curtail future damages across all sectors.

Introduction

Biological invasions are a major component of anthropogenic global change, causing significant disruption to ecosystems across regions, habitat types and taxonomic groups (Bellard, Genovesi & Jeschke, 2016; Essl et al., 2020). Across taxa, the rates of biological invasion continue to increase globally (Seebens et al., 2017; Seebens et al., 2021), with impacts that challenge conservation efforts, management interventions and socioeconomic enterprises (Hulme, 2009; Early et al., 2016). Such impacts are most visible on islands where high levels of endemism, naïveté, elevated introduction rates, and often limited economic capacity to respond, mean invaders can have substantial impacts (Courchamp, Chapuis & Pascal, 2003; Bellard et al., 2017; Russell et al., 2017; Anton et al., 2020). The determination of ecological impacts of invasive alien species has progressed considerably in recent decades, with means of predicting impacts on native communities (Dick et al., 2017) and potential new invaders (Fournier et al., 2019), across multiple scales. However, despite generally widely known ecological impacts (but see Crystal-Ornelas & Lockwood, 2020), effects of biological invasions on socioeconomic sectors such as human health, fisheries or agriculture have lacked synthesis (Paini et al., 2016; Shackleton, Shackleton & Kull, 2019), and have not been integrated with advances that provide qualitative means to define impacts on human wellbeing (Bacher et al., 2018). Together this has reduced incentives for policy makers to respond to biological invasions, owing to a lack of monetary quantifications of invasive species impacts, despite methodological advances for quantifying benefits of invader management in economic terms (Bacher et al., 2018; Hanley & Roberts, 2019).

Invasive alien species (IAS) can impact economies in both conspicuous and inconspicuous ways. Particularly, impacts from invasions can encompass resource damages and losses (Paini et al., 2016), proactive and reactive spending on various forms of management to prevent, control and eradicate invaders (Robertson et al., 2020), and less direct environmental impacts that adversely affect, for example, tourism and recreational activities (Hanley & Roberts, 2019). For management interventions, expenditure in preventative biosecurity measures for invasions can be more cost efficient than longer term management (Leung et al., 2002; Ahmed et al., 2022) and can help to negate future damages. However, most national economies, i.e., the scale at which policy decisions are made, have no centralised, systematic or comprehensive means of reporting, nor collating, economic costs of biological invasions (Diagne et al., 2020). For example, means of identifying taxonomic groups that are most damaging, habitat types most impacted, and sectors most affected could help in efficiently directing allocations of resources (Cuthbert et al., 2021a; Cuthbert et al., 2021b). Previous highly-cited works on the economic impacts in the United States (Pimentel et al., 2001; Pimentel, Zuniga & Morrison, 2005) and Europe (Kettunen et al., 2009) brought valid attention to the burgeoning economic impacts of invasions, but they have come under scrutiny (e.g., Cuthbert et al., 2020) given unclear methodologies, such as the inability to distinguish which costs were empirically observed, and a reliance on sometimes extreme extrapolations from small, localised scales.

New Zealand comprises a large archipelago—centred on the North Island Te Ika-a-Māui (114,453 km2), the South Island Te Waipounamu (150,718 km2), and Stewart Island Rakiura (1,746 km2)—scattered throughout the southwest Pacific Ocean (latitude: 30°S to 52°S, longitude: 165°E to 175°W). New Zealand has been heavily affected by IAS and is widely regarded as a country at the forefront of IAS management (Hayden & Whyte, 2003; Russell et al., 2015; Simberloff, 2019). This country was one of the last land masses on Earth to be colonised by humans and, as an isolated archipelago, was, and is, typically vulnerable to the severe impacts of introduced species (Allen & Lee, 2006). Together, this meant early natural historians, and subsequently wildlife managers, simultaneously saw the impacts and costs of IAS, but also the benefits that could be gained from their successful management (Department of Conservation, 2020; Bellingham et al., 2010; Towns, West & Broome, 2013; Bell, Bell & Merton, 2016). This also meant, from c.a. 1900 onward, New Zealand had a strong degree of biosecurity to prevent unwanted introductions, originally with a focus on agricultural protectionism, but later expanded to include biodiversity protection (Hayden & Whyte, 2003; Hulme, 2020). The Biosecurity Act 1993 established strict IAS screening protocols in New Zealand, based on a rigorous evidence-based assessment of the risks associated with a given overseas port or on particular goods (Jay, Morad & Bell, 2003). While expensive (e.g., averaging $485 million NZD per year from 2017–2020; New Zealand Government, 2019), these protocols are implemented with the intention of avoiding greater future costs associated with established IAS management and damages (Leung et al., 2002; Ahmed et al., 2022). This is in contrast to the vast majority of countries globally where a blacklisting approach (i.e., only specifically named species are banned), and a greater reliance on individual responsibility to declare biological material, is typical. New Zealand has also led the world in pioneering the eradication of IAS—including established mammal species from small islands (Towns, West & Broome, 2013), incursions of arthropods on the large, inhabited islands (Brockerhoff et al., 2010) and the eradication of non-native plants (Hulme, 2020). One particular feature of New Zealand—the only remote archipelago that is a developed nation independent of any continental authority—positions this country well to be a world leader in IAS management (Simberloff, 2019). However, despite its successes and place at the forefront on the global stage, the country still suffers from invasion debts (i.e., the time-delay before an already introduced species becomes invasive; Sheppard, Burns & Stanley, 2016; Brandt et al., 2021) and incursions of new known IAS that pose substantial ongoing challenges to management.

The first global compilation of reported invasion costs (‘InvaCost’) has recently been published (Diagne et al., 2020; Diagne et al., 2021a), providing the most comprehensive, synthesised database of IAS economic impacts to date. It also provides a standardised living platform for the reporting, comparison and synthesis of invasion costs in the future, enabling continuous updating by scientists, managers and stakeholders (Diagne et al., 2020; Diagne et al., 2021a). While limitations remain, as awareness and uptake of this platform increases, it will be invaluable to IAS research. For example, a lack of easily accessible damage costs can impede efficient allocation of targeted management across species or habitats (Ahmed et al., 2022). Using data available from the InvaCost database, we summarised and described reported economic costs of invasions in New Zealand across approximately the last fifty years. Assessments of IAS in New Zealand already exist, but are typically small-scale or limited to specific economic sectors or taxonomic groups (Hackwell & Bertram, 1999; Barlow & Goldson, 2011; Allen & Lee, 2006; Nimmo-Bell, 2009; Clout, 2011; Saunders et al., 2013; Ferguson et al., 2019). Here, we investigated how the currently available reported costs of invasions are characterised across all (a) environments, (b) cost types, (c) economic sectors and (d) taxonomic groupings. We also examined the temporal trends in the costs of both damage to resources and management investment, where we predict an increase in damage costs through time given the ongoing increase in biological invasions worldwide (Seebens et al., 2017), but accompanied with a significant increase in management costs as a result of New Zealand’s broadly proactive approach to managing IAS impacts (Towns, West & Broome, 2013; Simberloff, 2019). We interpret these results within the specific context of New Zealand, and more broadly to the reported economic costs of invasions in other countries and regions internationally.

Methods

Data collection

To describe the cost of invasions in New Zealand, we used cost data collected in the latest available version of the InvaCost database (v4.0), as of June 2021 (Diagne et al., 2020; Angulo et al., 2021); data openly available at https://doi.org/10.6084/m9.figshare.12668570). Full details on the process of the literature search are provided elsewhere (Diagne et al., 2020). Briefly, three online bibliographic sources (Web of Science, Google Scholar, and the Google search engine) were examined using a series of carefully composed search strings, creating standardised searches within the peer-reviewed and grey literature for the economic costs of IAS. Such costs could be presented at any taxonomic level, spatial scale or time period and to any economic sector. All cost entries were standardised to a common and up-to-date currency (2017 US$), although here we also report them as 2017 NZ$ (US$ ×1.4 as of 1 Jan 2017) for national context. The official market exchange rates were obtained from the World Bank Open Data and adjusted using an inflation factor that accounted for the changes in US$ since the cost estimation year using consumer price indices (Diagne et al., 2020). Using the “Official_country” column within the database, we filtered entries for New Zealand, including all reported costs from 1960–2020 to ensure accurate cost standardisation and the latest year from which costs have been collected respectively (n = 2 pre-date this point [1883 and 1945]). We report results as both cost values and number of reports—the latter reflecting a standardised measure (i.e., costs per year) of cost reporting effort for a given variable. We note that the searches made to compile the database may under-represent groups such as microbes (e.g., pathogenic bacteria or viruses) because, although their spread can be dramatically enhanced by invasive hosts, they are frequently not defined by authors as IAS in their own right (Vilcinskas, 2015; Roy et al., 2017). The subset of the database used for New Zealand is provided as Supplementary Material 1.

Estimating total costs

Deriving the cumulative cost of invasions over time requires consideration of the duration of each cost occurrence. We calculated this cost duration as the number of years between the database columns “Probable_starting_year_adjusted” and “Probable_ending_year_adjusted”. In a few instances, costs were omitted (n = 12) because they had unclear durations because either the start, end or both years were not specified in the original source. We opted to remove these entries to avoid biases when assessing temporal dynamics of costs (i.e., with costs specified over too few or many years). To calculate the total cumulative cost, we first standardised all the cost entries on an annual basis for their defined period of occurrence. Hence, for example, a single cost entry recorded as occurring over a six-year period was transformed to six cost entries, with the total cost divided to get an annual cost that was repeated for each of the six entries. Our dataset therefore initially consisted of 810 annualised cost entries from 124 unique references.

Invasion costs were then considered and estimated according to six descriptive columns present in the database (see Diagne et al., 2020 and https://doi.org/10.6084/m9.figshare.12668570 for complete details on these descriptors):

(i) “Method_reliability”: a conservative but objective evaluation of the traceability of the cost estimation (“High” vs “Low”). This is based on the type of publication and method of estimation i.e., peer-reviewed or other official documents from the grey literature are likely validated prior to publication so are classified as “High” reliability. Other materials were only classified as “High” if the original sources, assumptions, and methods were accessible and fully described.

(ii) “Implementation”: referring to whether the cost estimate was actually realised in the invaded habitat (“Observed”) or whether it was extrapolated (“Potential”).

(iii) “Environment_IAS”: whether the cost was incurred from biota that are either “Aquatic”, “Terrestrial” or “Diverse/Unspecified” (i.e., either unspecified or a combination of species and hence biomes).

(iv) “Type_of_cost_merged”: collation of costs according to principal categories: (a) “Damage”, referring to damages or losses incurred by invasion (e.g., costs for damage repair, resource losses); (b) “Management”, comprising management-related expenditure (e.g., monitoring, prevention, control, eradication); and (c) “Mixed” costs, including a mixture of damage and management costs. We also used the “Management_type” column to compare management expenditure between pre- and post-invasion actions. Here, pre-invasion management comprised monetary investments for preventing successful invasions in an area including quarantine or border inspection, risk analyses, biosecurity management etc.; and post-invasion management includes money spent on managing invaded areas, and so includes control, eradication, or containment. Additional categories comprised: (a) “knowledge/funding”—money allocated to any action or operation that could be relevant to management at pre- or post-invasion stages, but is not specifically attributed within the source e.g., administration, communication, education or research costs;  (b) “mixed”—costs that included at least (and without possibility to disentangle the specific proportion of) two of the previous categories; and (c) “unspecified”—costs where the exact nature was not clearly defined. 

(v) “Impacted_sector”: the activity, societal or market sector that was impacted by the cost (e.g., “Agriculture”, “Health” or “Authorities and Stakeholders”—this latter category representing official structures and organisations allocating efforts to manage IAS). Individual cost entries not allocated to a single sector were classified as “Mixed”, and records without an identifiable sector, or those that were unreported, were classified as ”Unspecified”. These are relatively broad groupings as the level of granularity provided within references varied substantially. Importantly, there was very rarely sufficient detail to attribute costs to, for example, specific stakeholders, communities or specific health impacts, even if these were mentioned, rather than to the collective sector groupings detailed above.

(vi) “Species”: the taxonomic nomenclature of the species causing the cost.

We focussed the majority of our analyses on a conservative approach to the reported costs, retaining only those which were graded as both observed (i.e., had actually occurred) and having ‘high’ reliability (i.e., a level of peer review or method reproducibility), hereafter called ‘robust’ costs.

To analyse the economic costs of IAS over time, we used the summarizeCosts function in the R package “invacost” (Leroy et al., 2022). With this function, we calculated (i) the average annual costs over the entire 1960–2020 period (in 2017 US$), and (ii) the total costs per decade. This temporal analysis was performed separately for management and damage costs. The InvaCost database includes data covering a range of spatial scales from site-specific to national level. We considered all scales in our analysis as there is no reason to expect that cost amounts and spatial scales show a linear relationship (e.g., some impacted items may be very expensive despite small impacted areas, others may benefit from economies of scale), but carefully checked the data for duplicates across scales, e.g., the same species-location pair reported twice. Identified duplicates were eliminated. We did not extrapolate cost estimates from smaller to larger scales, and such extrapolations are only included in InvaCost if the underlying cost documents did so and explicitly described their estimation methods (see “Implementation” above). In many cases, a lack of extrapolation could render our results underestimates, but we stress that InvaCost is a compilation of reported costs from underlying studies, that extrapolations can come from either “High” or “Low” reliability sources (see above), and that such extrapolations from smaller scales have been previously recognised as potentially problematic (Cuthbert et al., 2020).

Finally, we used natural log–log linear regressions to examine the relationship between Gross Domestic Product (GDP) and total (a) pre-invasion management (b) post-invasion management and (c) damage costs for New Zealand, as well as for all other countries represented in the InvaCost database (using the same filtering system to include only highly reliable, observed costs between 1960 and 2020 with specified temporal durations in each case). Accordingly, three linear models were fit with each of these log-transformed cost types as a response variable, and with log-transformed GDP as an explanatory variable per country. From these we extracted the ratio of [observed over expected] expenditures within a country for pre-invasion and post-invasion management, i.e., the residuals of the GDP/expenditure correlation. We used these relationships to highlight differences in New Zealand’s expenditure in comparison to global trends, relative to economic output.

Results

From the InvaCost database we obtained 124 unique cost records encompassing a minimum of 52 species (a number of entries did not define specific species in their assessments, see Table S1 for a full list) that corresponded to 810 annualised cost entries between 1968–2020, summing to a total reported value over this period of US$69.09 billion (NZ$96.73 billion). Over half of the cost entries (n = 453) were directly observed from actual costs, with the remainder being predicted, potential costs. Of the directly observed cost entries, over four fifths (n = 368) were classed as highly reliable, with a total figure for these highly reliable, observed costs of US$8.83 billion (NZ$12.36 billion, Fig. 1). The following analyses focus only on these highly reliable, observed costs, hereafter called ‘robust’ costs.

Figure 1 The economic cost of biological invasions in New Zealand from 1960 to 2020 from the InvaCost v4.0 database using only robust costs.

Costs are totalled according to the environmental type in which they occurred and, within each environment, by the socioeconomic sectors impacted and the type of costs (management or damage) incurred. Only sectors with costs greater than US$50,000 are presented for clarity. Widths for type and sector costs are scaled relative to their environmental cost contributions. Environmental costs are labelled in white within the figure scene.

For these robust costs of biological invasions, approximately one third of the total cost (36%), and most annualised entries (62%), occurred within the terrestrial environment (Fig. 1, Table S2). Cost entries were common from the aquatic environment (29%), but with a very low proportion of the total recorded cost (1%), with the remainder from diverse/unspecified environments (9% of entries but 62% of cost, Fig. 1, Table S2). The majority of recorded cost entries (65%) were associated with management; damage cost entries represented a much smaller proportion (35%), but management investments were far smaller than damage cost (18% vs. 82%) (Fig. 1). The greatest proportion of the robust cost total was borne by the primary industries of agriculture and forestry (47% combined), and this sum represented 27% of the annualised entries. Other significant proportions of the robust cost total were borne by unspecified sectors (36%)—although this reflected a very small number of annualised entries (<1%)—and Authorities and Stakeholders (16% of total cost but 50% of entries). There was very limited total cost reported from the health, public and social welfare, and environment sectors (all <1%) (Fig. 1, Table S3). The majority of cost entries were caused by animals (88%; 15% of cost), then plants (8%; 31% of cost) and chromists (2%; <1% of cost); with diverse or unspecified groups contributing the remainder of entries and the majority of the total cost.

Estimates of annual management costs since the 1960s averaged at US$26.16 million while damage costs were an order of magnitude higher at US$120.41 million; management costs were first reported a decade later than damages (Fig. 2). Despite undulations in recent decades, management and damage costs both tended to increase through time, with management costs peaking at US$122.72 million per annum in the 2000s, and damage costs peaking at US$488.40 million in the 2010s. However, damage costs were consistently greater than management costs within each decade (Fig. 2). In general, numbers of costs reported increased over time considering the robust data (Fig. 2). Across all environments, management costs have most consistently been spent on control, although expenditure on other management types has increased by one order of magnitude in the most recent decade (Fig. 3), with eradications representing the greatest total spend (Table S4). As a result, post-invasion management spending (US$1.41 billion) massively exceeded pre-invasion management (US$0.07 billion) (Table S4).

Figure 2 Temporal development of management and damage costs of biological invasions in New Zealand from the InvaCost database (v4.0) using only robust costs.

Each respective orange and green dot represents the annual reported costs for management and damage (no dot means an absence of reported costs for that year). Solid horizontal bars and squares indicate decadal averages of economic costs (management: orange; damage: green), with fine dashed lines linking the decadal means.

Figure 3 Heatmap demonstrating the magnitude of costs (US$ billions) of IAS in New Zealand as a function of management cost type per decade across differing environments.

Colourless sections indicate that no cost values have been reported.

Robust costs included information on a minimum of 52 IAS, with those reported as incurring the greatest damage costs (>US$200 million) being agricultural weeds (e.g., creeping thistle (Cirsium arvense), yellow foxtail (Setaria pumila), giant buttercup (Ranunculus acris)) and pest arthropods (e.g., Argentine stem weevil (Listronotus bonariensis)). Introduced species reported as incurring the greatest management costs (>US$50 million) were also associated with primary industries but were a different group of species that included the established pest animal brushtail possum (Trichosurus vulpecula) and the Varroa mite (Varroa destructor). Noteworthy management costs included those to thwart the establishment of invading pest arthropods (e.g., painted apple moth (Teia anartoides), red imported fire ant (Solenopsis invicta) and gypsy moth (Lymantria dispar)) and eradications of established pest mammals on isolated islands (e.g., feral cats (Felis catus) and European rabbits (Oryctolagus cuniculus)) (see Table S1 for a full list of specific species costs).

Invasion costs related significantly positively to GDP considering damage (t = 5.688, p < 0.001) and post-invasion management (t = 6.549, p < 0.001), but not pre-invasion management (t = 1.604, p = 0.123), at the global scale. In terms of these global economic cost patterns, reported costs across both management types are relatively high in New Zealand, and higher than would be predicted based on GDP (Fig. 4, Fig. S1); although this likely reflects study effort to some extent and not solely economic impact. Based on the global pattern among the 20 countries reporting both management types, post-invasion and pre-invasion management in New Zealand were both about 5 times higher than expected based on GDP. However, globally, pre-invasion spending is less frequently reported overall (Fig. 4, Fig. S1).

Figure 4 Management expenditures related to expectations from correlation with GDP for all countries within the InvaCost database where pre- and post-invasion management approaches are recorded.

Each point represents the ratio of [observed over expected] expenditures within a country for pre-invasion and post-invasion management. The area shaded blue contains all countries with lower than expected expenditures (ratio <1) for either pre- or post-invasion management. New Zealand (red dot) spends significantly more than expected from its GDP on both pre- and post-invasion management costs.

Discussion

Our results provide robust evidence that biological invasions incur substantial and increasing economic costs through diverse negative impacts to socio-ecosystems. In New Zealand, costs have reached at least US$8.8 billion (NZ$12.4 billion) over the last 50 years based on only the most robust estimates. Most costs have resulted from damage caused by terrestrial IAS, and costs are rapidly increasing over time. Despite New Zealand’s well-deserved reputation as a world leader in addressing IAS from a variety of pre- and post-invasion perspectives, this situation where damage costs are growing and substantially exceeding management investments is similar to many other countries and regions (e.g., Bradshaw et al., 2021; Haubrock et al., 2021) and confirms our hypothesis. Previously, it has been estimated that IAS are regularly associated with over NZ$1 billion a year of losses in New Zealand (Goldson et al., 2015), corresponding to over 1% of the national GDP (Nimmo-Bell, 2009). We show here that reported economic costs, which are conservative estimates given that robust values are not present for all species (Bradshaw et al., 2016; Hoffmann & Broadhurst, 2016; Diagne et al., 2020), are approaching this value. Thus, inclusion of additional information such as predicted cost estimates, would result in this value being exceeded over the past decade. Our comparison of robust direct economic costs across different invasive taxa and invaded ecosystems provides context to the level of economic threat biological invasions pose, and a minimum damage value to weigh against response and mitigation expenditure (Turner et al., 2004; Dasgupta, 2021).

Damage costs made up the principal economic burden and were mostly reported from the terrestrial primary industries that dominate New Zealand’s economy (NZ$38.1 billion export value in 2017; Ministry for Primary Industries, 2017), and where such costs are most easily estimated through direct financial losses. Management costs also focused, in part, on remediating this damage. However, this spend focussed not just on introduced species currently incurring large damage costs, but also on preventative measures where upfront investment in management would prevent large future damage costs being incurred. While reported post-invasion management expenditure overwhelmingly predominated over pre-invasion expenditure, costs associated with the latter—e.g., in terms of biosecurity staff salaries, facility costs, research grant breakdowns etc.—are rarely within the public domain (for example as a result of privacy legislation) and thus are unlikely to be captured within this database. This makes a completely accurate comparison between proactive biosecurity management (i.e., that undertaken at early invasion stages) and post-invasion control/eradication challenging due to data deficiencies. However, New Zealand now invests a relatively high amount in IAS prevention activities such as pathway management border controls, and is a world leader in pre-border pest risk management (Brenton-Rule, Frankel & Lester, 2016; Hulme, 2020). These prevention activities often apply to multiple species simultaneously, so they are anticipated to be a more cost-effective way of achieving the same benefits as management activities further along the invasion spectrum (Leung et al., 2002; Ahmed et al., 2022). However, empirical tests of this comparison are complex and ethically challenging, requiring a robust design applying multiple management approaches (pre- vs. post-invasion at a minimum) to the same invader and ecosystem across comparable time periods. Nevertheless, due to a long history of, and ongoing dependency on, primary production, and more recently tourism, New Zealand continues to emphasise a proactive approach to biosecurity, which has benefits for multiple industries (Jay, Morad & Bell, 2003; Brockerhoff et al., 2010).

As well as a focus on mitigating economic damage to industries, substantial investment in New Zealand has also been made in managing IAS threats to biodiversity, particularly the initially high up-front investment in eradication, with its anticipated long-term reduction in economic or ecological costs (Bomford & O’Brien, 1995). This can be seen in the significant proportion of post-management expenditure on this common approach for introduced mammals on uninhabited offshore and “mainland islands” of New Zealand (Carter et al., 2021); and has now been proposed nationally (Department of Conservation, 2020; Russell et al., 2015). Managing threats to biodiversity often occurs despite a lack of detailed information on the economic impacts of such actions and, indeed, many benefits associated with biodiversity and healthy ecosystems are not easily quantified monetarily. Indeed, some impacts—for example spiritual and cultural costs—are arguably impossible to value purely economically, although approaches such as revealed preferences can be used (Shackleton et al., 2019a; Shackleton, Shackleton & Kull, 2019). Such differences potentially skew economic assessments towards species which cause impacts that are readily monetized, and implicitly bias cost reporting against those which primarily impact native biodiversity and ecosystem functioning. However, despite debate around attributing economic costs, the loss of ecosystem services—e.g., carbon sequestration, recreation, or culturally significant sites—can play a key role in justifying IAS management (Holmes et al., 2009; Hanley & Roberts, 2019).

Regardless of the types of impact, management of biological invasions require additional economic spending (Clout, 2011). However, in New Zealand, decadal means for reported management costs (including all costs identified under control, eradication, biosecurity and research/communication categories) have been exceeded by decadal means for reported damage costs for the past forty years. This is also true across the majority of recent years (Fig. 2), with a potentially worrying increase in the gap between management spend and damage incurred in the most recent decade (although this is necessarily caveated by incomplete reporting for the most recent years given inevitable publication lags (mean = 6 years)). This situation is reflected in many parts of the world where economic losses from damage far outstrip management costs (Crystal-Ornelas et al., 2021; Diagne et al., 2021b; Haubrock et al., 2021; Liu et al., 2021). Although the overall management spend in New Zealand does not appear sufficient to maintain pace with reported damages, the specific focus of investment into protecting taonga (i.e., ‘treasure’—culturally significant species that shape Mātauranga Māori (Māori knowledge)); (Collier-Robinson et al., 2019), often extremely successfully (Allen & Lee, 2006; Bellingham et al., 2010; Towns, West & Broome, 2013), with their less readily monetised benefits may also explain some of this difference. This approach may even increase in the coming decades with the Predator Free 2050 program (Department of Conservation, 2020), although this also has clear long-term objectives to reduce both management and damage related expenditure. Indeed, when disentangling pre-invasion from post-invasion management in New Zealand, proportional spending on early-stage management was considerably higher than in many other countries within the InvaCost database when considering their economic output. This stresses the relative scale of funding allocation to early-stage investments in New Zealand, and highlights the potential for further gains.

In addition to differential reporting of ecological costs, there are other costs of biological invasions missing from the database for a range of reasons. There are undoubtedly true gaps in reported damage costs, and consequent lack of, or limited, investment in management. These include introduced game animals e.g., chamois (Rupicapra rupicapra) and common pheasant (Phasisanus colchicus) and freshwater fish such as trout (Salmo trutta), which engender conflict through opposing values on whether they are pests (thereby incurring a damage cost) or resources (thereby not requiring control and, indeed, providing economic benefits) (Russell, 2014). This conflict likely results in absences of spending on management, combined with controversy and a reluctance to attribute damage costs, although they do still controversially occur (Hughey & Hickling, 2006). Economic assessments are similarly lacking or speculative for many parasites and diseases as a result of debate over their description as IAS in their own right, their cryptogenic nature, and a generally poor understanding of their epidemiology or virulence prior to emergence (Gross et al., 2014; Vilcinskas, 2015; Roy et al., 2017). This is true even for those that damage commercially-important species, for example Bonamia ostreae which can adversely affect oysters (Ostrea chilensis) (Lane, Webb & Duncan, 2016). Reported harm to human health was also low, and is likely underestimated (Wilson et al., 2018); although this may still reflect an actual low level of impact from biological invasions on human health in New Zealand. Lastly, species such as clover root weevil (Sitona lepidus), pea weevil (Bruchus pisorum) and termites (Order Isoptera) have had large amounts spent on them by the Ministry of Primary Industries and Biosecurity New Zealand, but this information remains in internal reports and spreadsheets that have yet to leave the organisation (S. Wood, pers. comm. August 31, 2020). This level of difficulty in accessing relevant data, and the inability of searches to locate such information without the need to specifically target key, potentially unknown, individuals or researchers, represents a major barrier to obtaining accurate pictures of the economic impacts of IAS. Therefore, parties that collect these data, including central government, regional agencies and scientists, should also prioritize their publication such that their availability and impact are maximized, facilitating, for example, independent advice to governments or the linking of environmental management to broader concepts of wellbeing.

Emerging threats in biological invasions, and a wider acknowledgement that costs are simultaneously incurred, and borne by, multiple sectors, has led to greater cross-agency partnership in the management of biological invasions in New Zealand (Jay, Morad & Bell, 2003). However, managing such multi-party responses brings additional complexity with respect to ensuring sufficient resource allocation and effective inclusion of cultural values and viewpoints, particularly when significant expenditure is at stake (Ngā Rākau Taketake, 2019). It also has the potential to further complicate open, accessible reporting of costs if lines of responsibility are unclear. A current ongoing example of multi-agency operations, given the invasion process is identical and responsive to the same attitudes and policies as unwanted organism management, is New Zealand’s rigorous biosecurity-focussed approach to SARS-CoV-2 (Nuñez, Pauchard & Ricciardi, 2020), where $74.1 billion NZD has been allocated in support of response and recovery initiatives (New Zealand Government, 2022). Given the ongoing potential for IAS to facilitate emerging infectious diseases, such proactive and integrated approaches would be highly beneficial (Hulme, 2014; Roy et al., 2017; Ogden et al., 2019). At the global level, reactive management towards IAS has been shown to outweigh proactive approaches 25-fold, with timelier management having the potential to save multi-trillion dollars in avoided impacts (Cuthbert et al., 2022). Total eradication of unwanted organisms has the potential to contribute widely to all aspects of society and its sustainable development (de Wit et al., 2020) and this view is readily accepted in New Zealand, where eradication of IAS is regularly pursued whenever technically possible and economically reasonable (Department of Conservation, 2021).

The information contained in the unique source of cost data analysed here provides a basis for helping to guide and encourage open reporting practice to facilitate research, management, and active steps to address the socio-economic impacts of IAS in New Zealand and beyond. We again stress that knowledge gaps and inaccessibility of potentially incurred costs make the costs reported here substantial underestimates. Nonetheless, as a centralised platform for standardised cost reporting, we encourage researchers and organisations to submit their data to the InvaCost database to enhance the comprehensiveness of future cost appraisals as biological invasions, their impacts and mitigation measures ensue. This will ensure a more complete understanding of economic costs and benefits under different strategies, a critical requirement for successful prevention and management of biological invasions.

Supplemental Information

Supplemental Information 1 Abstract (in Maori)

Click here for additional data file.

Supplemental Information 2 Supplementary Tables and Figures

Click here for additional data file.

Supplemental Information 3 InvaCost v4 Dataset for New Zealand used in this study

Click here for additional data file.

Additional Information and Declarations

Competing Interests

Author Contributions

Data Availability

The authors declare there are no competing interests.

Thomas W. Bodey conceived and designed the experiments, performed the experiments, analyzed the data, prepared figures and/or tables, authored or reviewed drafts of the article, and approved the final draft.

Zachary T. Carter conceived and designed the experiments, performed the experiments, analyzed the data, prepared figures and/or tables, authored or reviewed drafts of the article, and approved the final draft.

Phillip J. Haubrock conceived and designed the experiments, performed the experiments, analyzed the data, prepared figures and/or tables, authored or reviewed drafts of the article, and approved the final draft.

Ross N. Cuthbert conceived and designed the experiments, performed the experiments, analyzed the data, prepared figures and/or tables, authored or reviewed drafts of the article, and approved the final draft.

Melissa J. Welsh conceived and designed the experiments, authored or reviewed drafts of the article, and approved the final draft.

Christophe Diagne conceived and designed the experiments, performed the experiments, authored or reviewed drafts of the article, and approved the final draft.

Franck Courchamp conceived and designed the experiments, performed the experiments, analyzed the data, authored or reviewed drafts of the article, and approved the final draft.

The following information was supplied regarding data availability:

The raw data are available in the Supplementary Files.

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
