# Peer review of "Building a synthesis of economic costs of biological invasions in New Zealand"

_PeerJ, doi:10.7717/peerj.13580_

## Round 0.1 · original submission · Minor Revisions

Reviewers' comments on your work have now been received. The manuscript has been assessed by three reviewers. Reviewers indicated that the introduction, methods, results, discussion and references should be improved. Moreover, the figures should be provided. I agree with this evaluation and I would, therefore, request for the manuscript to be revised accordingly.

Reviewer 2 has requested that you cite specific references. You may add them if you believe they are especially relevant. However, I do not expect you to include these citations, and if you do not include them, this will not influence my decision.

Reviewer 1 ·

Basic reporting

My comments are in given in detail in the below section on "additional comments".

Experimental design

My comments are in given in detail in the below section on "additional comments".

Validity of the findings

My comments are in given in detail in the below section on "additional comments".

Additional comments

This manuscript represents an economic assessment of invasive species for New Zealand over the period of 1968- 2020. Economic assessments are notoriously difficult to achieve, as they are full of unreported costs and assumptions. But the authors do a good job using a newly created database and acknowledge limitations in the analysis. The manuscript needs a careful and thorough review and tidy-up for the grammar and presentation in main manuscript, the figures, and the references. I also have some points for the authors to consider below. After revision I think the paper should be acceptable and would make a useful addition to the literature that would be well cited. The points to consider are:

Abstract. The abstract is fine, although doesn’t mention the multi-country analysis that you show on figure 4.

Introduction. The introduction is long and extensive, covering much of the relevant literature. You might want a sentence or two introducing NZ to readers that don’t know much about NZ: where it is, the size, and perhaps the fact that we have suffered extensively from biological invasions including a lot of extinctions. On line 38, I don’t think you need “(but see Crystal-Ornelas and Lockwood, 2020)”. Just say “however, despite GENERALLY widely known…”.

Methods. I thought the methods were presented in sufficient detail to enable the reader to understand what is happening. My only suggestion would be to expand on what the log-log-linear analysis is testing and how to interpret that analysis. As I note in the section below on the references, these need updating. For example in the methods you state tha Leroy et al. (2020) just on bioRxiv. Has it not moved on from bioRxiv yet?

Results. I liked the results and the focus on the “highly reliable costs”. On lines 234- 235 you state that there has been a very limited total cost reported from health, public and social welfare sectors. It would be useful to follow that up in the discussion or additional analysis. My suspicion is that the invasive species we have in NZ haven’t had a major health or social welfare impact here. Note on lines 258 that NZ has “successfully” eradicated many of the invasive species you mention.

Discussion. The discussion is very long. As it stands I think it’s length dilutes your primary messages. Perhaps paragraphs such as lines 390- 404 could be removed. Despite and in contradiction of my comment on the length of the discussion, there are a couple small additions that you could consider. I don’t think your definition of taonga is quite there, as it is often considered ‘treasure’. In your discussion on management costs, you could talk about species such as rabbits, given they are an increasing cost because the government has put less money into managing them (no more rabbit boards, etc). The discussion on SARS-CoV-2 is interesting as it could well be considered an “invasive species”. How many billions has that cost us? The discussion is certainly an area that the punctuation and grammar needs work. It also has some VERY long paragraphs.

References. The references need careful inspection and revision.The first two references (Ahmed et al. 2021 and Allen et al. 2006), for example, are both incomplete. Please include page numbers issues, titles, etc.

Figure 1 is a nice idea and start, but needs more work. What are the units on the y-axis? Billions of NZ or US dollars? The units aren’t stated in the caption or on the graph. Please ensure the axis labels are in the right place.

Figure 2. Does this graph show the costs are leveling-off over time, or is that a function of the logarithmic scale? The horizontal error bars are weird. They are suggested to represent the variation around the decadal averages of economic costs, but the horizontal or x-axis is years. I think this graph presentation needs further thought.

Figure 3. The authors need to make it clear in the caption what the white sections of the plots are, which I assume to represent a lack of information.

Figure 4. This figure and analysis is not mentioned in the abstract. More detail is needed for the axis units. The areas of the graph highlighted in blue are the countries suggested to have “lower expenditures than expected by GDP”. How did the authors derived these expectations? Such a derivation is not mentioned in the methods and is highly subjective. I suggest removing it.

I hope these comments are useful for improving the manuscript.

·

Basic reporting

The manuscript by Bodey et al. reviews the costs of biological invasions in New Zealand. This study is part of a larger global initiative, INVACOST, that is providing assessments across nations, taxonomic groups and socio-economic sectors. This is important to note, because the value of the study must be considered in this context: results are not only interesting by themselves, but could be compared with the rest of assessments published or in the process of being published.

The language, structure and figures of the paper are clear and the manuscript is easy to follow. The sources of data are indicated and readers are encouraged to even provide new evidence to this growing database.

Experimental design

The research question is well defined and meaningful. Indeed, New Zealand is a global reference for good practices in invasive species research and management. It is therefore important to quantify, and compare with other countries, the costs and benefits associated to management. Methods are well explained and replicates the statistical workflow of other Invacost papers, which should facilitate comparison.

Validity of the findings

The costs of biological invasions in New Zealand are estimated in N>$97 billion over the last 50 years. More than 80% correspond to damage costs. Yet, in comparison with other countries, New Zealand is investing considerably more on management.

I only have one criticism. The manuscript misses an important opportunity to quantify the economic benefits (or avoided damage) associated to the strong regulation of IAS in New Zealand, which is currently stated but not fully quantified. First, it is not clear when did regulations start. Even considering delays in the implementation of measures, it’d be very important to identify in temporal trends (e.g. Fig. 2) such important moments. Second, other Invacost papers have used methods to estimate the “avoided cost” of managing invasions (see Ahmed et al. 2021 https://orcid.org/0000-0002-8402-5111; Cuthbert et al. in press in STOTEN https://doi.org/10.1016/j.scitotenv.2022.153404). In spite of the limitations that the method may have, this seems more interesting than the simple comparison with other countries provided here, which is similarly subject to many limitations in data availability. Third, while I understand the use of this decadal time analysis (the one in Fig. 2), I wonder if other statistical approaches wouldn’t be more appropriate. For instance, piece wise regression analysis to identify potential inflexion points.

Additional comments

Page 8, line 71. “relatively early”. Can you be more specific?
Page 9, lines 73-75. “New Zealand has strict IAS screening protocols based on a rigorous evidence-based assessment of the risks associated with a given overseas port or on particular goods”. Since when exactly? Can you be more specific?
Page 9, line 75. How expensive? Can you be more specific?
Page 129, line 129. Can you explain the relevance of the number of records? Why would this information be as important as the total amount invested?
Fig. 1 could be nice as graphical abstract, but not very informative.
Fig. 2 could indicate the start of major IAS regulations.
Fig. 3 is confusing and little informative. Can you think of a different way to represent this information?

---

## Round 0.2 · accepted · Accept

The authors have improved their work based on the comments in the previous round of review.

Reviewer 1 ·

Basic reporting

Nothing more to add from the first review.

Experimental design

Nothing more to add from the first review.

Validity of the findings

Nothing more to add from the first review.

Additional comments

I’ve now read the revised manuscript and the author’s response to the referee’s comments. I’m now happy to recommend that the paper be accepted into the journal and have no additional comments. The authors have suggested the inclusion of an abstract in Te Reo or the Māori language, which I’d encourage the journal to allow. My only other suggestion would be to ensure the journal issue and page numbers are added at the proofing stage for the manuscript—there are several that need these if they are fully published in time.